# Anomalous Long-Range Attraction in Colloidal Binary Mixtures at Fluid–Fluid Interfaces

**Zonglin Yi and To Ngai ***

Department of Chemistry, the Chinese University of Hong Kong, Shatin, N.T., Hong Kong, China;
zonglinyibetty@gmail.com
\* Correspondence: tongai@cuhk.edu.hk

**Abstract:** The properties of binary colloidal systems have gained the interest of researchers because they have much richer structures than their one-component counterpart. Continuing efforts are being made on the theoretical side on binary colloidal systems, while many issues remained unsolved for the lack of solid experimental supports, especially for study in the field of two-dimensional (2D) binary colloids system. Oil–water interfaces can serve as a good stringent 2D confinement for colloidal particles and can avoid anomalous problems caused by the quasi-two-dimensional environment in previous experimental reports. In this work, we conduct experimental research of binary colloids system in an oil–water interface to revisit theoretical predication. We measure an ultra-long-range attraction and discuss the possible mechanism of this attraction by comparing the experimental result with existing model and theory. This study could contribute more understanding of the binary colloidal system in both experimental aspects and theoretical aspects.

**Keywords:** binary colloids; interaction between colloids; two-dimensional; oil-water interface

## 1. Introduction

Colloidal particles have garnered considerable interest given their good performance in both fundamental study and application-related material study. From a practical perspective, the self-assembly of colloidal particles into two-dimensional (2D) or three-dimensional (3D) colloidal crystal offers a bottom-up approach to fabricating functional materials such as photonic bandgap crystals, plasmonic materials, and nanowire arrays [1–4]. In terms of fundamental research, researchers have long used colloidal particles as an experimental model system to study phase transitions and dynamics in condensed matter physics since the 1940s, the time commercial latex became available. In comparison with atomic crystals where atoms interact through complex and predetermined potentials, colloidal particles can often be described by simpler interaction potentials, which can be more readily controlled at the particle level or via external forces. In addition, their easy observability by conventional optical microscopy and an instrumentally detectable time duration make them an ideal model system to fundamentally study the self-organization phenomena [5].

Studies using colloidal particles to test theoretical models for 3D system properties have been widely reported in the literature, including the study of freezing transition [6], glass transition [7], and jamming transition [8]. For example, Kose et al.'s experiment with monodisperse latex [6] substantiated the hard sphere liquid–solid Kirkwood-Alder transition, which had been theoretically studied and discussed for over 40 years [9,10]. Besides systems of only one colloid species (unary system), systems of different colloid species such as two-component system (binary system) also attract researchers, in which particles with different characteristics offer the system more versatility. For example, a system with different-sized particles sizes would have higher fluidity than a

single-component system [11]. In addition, mixtures of binary species demonstrated richer phase behaviors and structures than a single-species system, such as the formation of superlattices [12–14].

Theoretical studies on binary hard sphere systems [15] and binary hard disk systems [16,17] have been reported, with many problems remaining to be solved. One such example is about the stability of binary hard sphere system. An early study predicts that the binary system mixture is always stable regardless of the size ratio of the two species [18], while work by Biben et al. [15,19] via integral equation theory showed a contradictory result, that at more extreme size ratio, de-mixing happens in hard sphere binary system. In binary systems of charged particles, the phase separation issue is even more complex from theoretical prediction, because more parameters apply than in single-component hard sphere systems [20–22]. However, experimental studies presenting a good 2D reference model for binary particle systems are highly limited. Rice et al. [23] confined silica particles and polystyrene particles with two parallel plates to build a quasi-two-dimensional system and then used video microscopy to study structural properties and dynamics of the binary particle system. They found that the structure feature of large particles and interaction between them were closely related to the separation between two confining walls, indicating that the confining extent of set-up plays a role in experimental result output. A previous study [24] also revealed an enhanced attraction in a quasi-2D confined binary system, but the mechanism was unknown. Separation between two confining plates cannot create a strict 2D system, complicating the link between experimental results and theory. By contrast, trapping colloidal particles by interfacial tension at the oil–water interface provides a more stringent confinement to create a 2D system.

Therefore, in this work, we conducted experimental research of binary colloids system to revisit theoretical predictions in a stricter two-dimensional confinement: at the oil–water interface. Previously, we measured interactions between charged colloidal particles at the oil–water interface [25]. Based on that, we first created a colloid monolayer at oil–water interface using large 1 or 2 μm polystyrene (PS) particles. The behavior of the interfacial system was observed by microscope and we analyzed the structure of the interfacial colloid monolayer by calculating radial distribution function, *g(r)*, of the interfacial colloidal particles. Then we extracted the potential of mean force between the interfacial large particles in two-dimensional oil–water interface and qualitatively compared it with the reported quasi-two-dimensional results [24]. We then discussed in detail the effects of density of small particles (0.21 μm PS) to the binary system property and compare with the existing models. We measured an anomalous long-range attraction in interfacial binary particles, which is stronger than the three-dimensional system and the theoretical prediction.

## 2. Materials and Methods

### 2.1. Materials and Interfacial System Preparation

In this work, *n*-decane and ultra-pure water (Millipore, resistivity of 18.2 MΩ cm) were used for the oil phase and water phase, respectively. Decane was filtered through an $Al_2O_3$ column to remove impurities. The colloidal particles used were 2 μm of sulfate polystyrene (PS) (Invitrogen[TM], S37500), 1 μm of sulfate PS (Invitrogen[TM], S37498), 1 μm of carboxylate-modified polystyrene (PS) (Invitrogen[TM], F8820), and 0.21 μm of sulfate PS (Invitrogen[TM], S37491). Unary system (single-particle system) contained measuring PS particles with size of 1 or 2 μm, and corresponding binary system contained extra 0.21-μm small PS particles. All PS colloidal particles were negatively charged. Colloidal particles were washed via a centrifuge and re-dispersing process three times. A 10 vol% isopropanol (IPA)-contained particle suspension was finally prepared. Decane and ultra-pure water were degassed by ultrasonics prior to preparation of the oil–water interface. The interfacial system of oil–water was built in a delicate glass cell as described in our previous paper (Figure 1) [25]. Typically, the cell was designed with a glass coverslip as the bottom, a plastic spacer of poly(ethylene terephthalate) (PET) of approximately 100 μm, and a shallow glass dish with a hole in the bottom. The three parts were glued together with silica gel. The device was cleaned with plasma cleaner before interfacial colloidal system

preparation to provide the glass bottom with high hydrophilicity for water wetting. Ultra-pure water was first added. The PET spacer could pin the water at the edge, creating a flat planar water surface. The water layer was as thin as the spacer, facilitating ease of observation with high-magnification microscopy, which often has a short working distance. Then, an approximate volume of 2 mL of *n*-decane oil was gently added. Particle suspension was subsequently added to the interface to form an interfacial system. Often, a spreading solvent is involved as a common spreading method in experimental studies. After particle deposition, the device was double sealed by the inner glass and outer lid to prevent evaporation of the oil phase and environmental disturbance [25]. Interfacial system was kept overnight before data collection.

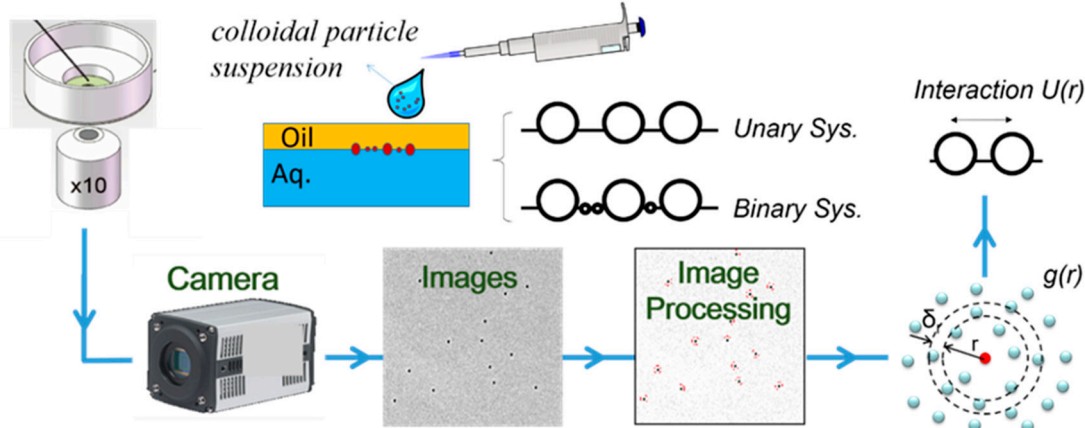

**Figure 1.** General method flow chart for measuring interaction potentials of colloidal monolayer and binary mixtures at the oil–water interface by recording images of interfacial particle monolayer, locating the particle via image processing, and calculating interactions between interfacial colloidal particles.

## 2.2. Data Collection

We took consecutive bright-field images of interfacial particles with an sCMOS camera (Andor Zyla) on an inverted microscope (Olympus IX-81) under 20× objective, where small 0.21-μm PS particles were invisible. A typical set of 2400 pictures with a frame rate of 2 fps were recorded. Small particles at the interface moved much more vigorously. Although they were visible with a higher-magnification objective, as the 0.21-μm particle size was nearly at the limit of optical microscope resolution, the image quality of time series photos with a faster frame rate was not useful in extracting reliable particle location information for small particles. Images with higher magnification substantially reduced the size of the view field. The time to take sufficient photos to ensure a statistically significant result would also be extremely long, leading to other problems with the reliability of the experimental results [25,26]. Thus, we only took snapshots with 60× objective magnification to show the structure of both large and small interfacial particles.

## 2.3. Data Processing

With a self-developed MATLAB image-processing algorithm, the locations of large particles were extracted, from which radial distribution function, $g(r)$, was calculated. Then, the pair-wise potentials $U(r)$ between particles was numerically evaluated from $g(r)$ by combining the hypernetted chain (HCN) approximation closure equation with the Ornstein-Zernike integral equation [25,26], in which $U(r) = -k_B T \ln[g(r)] + n k_B T I(r)$ was applied, where $r$ is the distance between two particles with the corresponding position $r_i$ and $r_j$ for particle $i$ and $j$, $k_B$ is the Boltzmann constant, $T$ is the absolute temperature. $I(r)$ is the convolution integral, $I(r) = \int [g(r')-1-nI(r)][g(|r'-r|)-1] \, d^2r'$, which can be solved iteratively by computer, starting with $I(r) = 0$.

## 3. Results

### 3.1. The Difference of g(r) in Binary and Unary Systems

Figure 1 shows our approach to measure the interfacial interactions between particles of the unary system or binary mixtures trapped at the water–oil interface. We first created a colloid monolayer at an interface using a customized device. Using video microscopy, we then took a series of images for interfacial particles. Next, we used MATLAB to process images and segregate particle locations. From the location information, we further calculated parameters to characterize system properties, such as the *g(r)* by reversible work theorem, from which we could derive the structure of interfacial system and extract the pair-wise potential (pair interaction) *U(r)* between interfacial particles. We then measured in detail about the effects of density of small particles to the binary system property and compared the calculated results with the existing models.

Figure 2 shows *g(r)* of large PS particles in unary and binary interfacial systems. The particle combinations of binary systems are: 2 µm sulfate PS with 0.21-µm sulfate PS, 1 µm carboxyl PS with 0.21-µm sulfate PS, and 1 µm sulfate PS with 0.21-µm sulfate PS. Since small and large particles are both negatively charged, they are repulsive. Intuitively, with the addition of small particles, the distance between large particles in a binary system should become longer due to the extra repulsion from small particles. To our surprise, large particles could approach each other at a closer distance in the presence of small particles, presumably due to some induced attractive force. This unexpected *g(r)* result warranted further exploration; therefore, we conducted a more systematic study with only charged PS of the sulfate surface group. The sizes of large and small particles were 1 µm (zeta potential −66.44 mV) and 0.21-µm (zeta potential −84.78 mV), respectively, and the zeta potential shows that both PS particles are negatively charged.

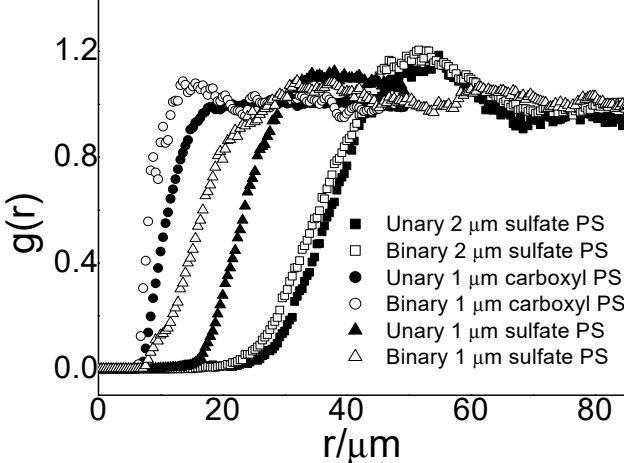

**Figure 2.** *g(r)* of large particles in presence and absence of small particles: large particles moved closer to each other in the binary system (hollow symbols) than in the unary system (solid symbols); ■ 2 µm sulfate polystyrene (PS) unary system; □ 2 µm sulfate PS + 0.21 µm of sulfate PS binary system; ● 1 µm carboxyl PS unary system; ○ 1 µm carboxyl PS + 0.21 µm sulfate PS binary system; ▲ 1 µm sulfate PS unary system; △ 1 µm sulfate PS + 0.21 µm of sulfate PS binary system.

It is worth mentioning that *g(r)* depends on the concentration of particles in the system. To identify the effect of small particle concentration, we kept concentrations of large particles almost the same in all test systems, and we used particle area fraction to characterize particle concentration. We controlled the area fractions $\phi$ ($= N\sigma^2/A$) of large particles to approximately $5 \times 10^{-4}$ in different systems to ensure no structural effects in this concentration level of large particles, where $N$ is the number of particles in an image, $A$ is the total area of the image, and $\sigma$ is the particle diameter. We compared the *g(r)* of

large particles under different small particle concentrations from roughly $10^{-4}$ to $10^{-2}$ and obtained the following results.

We use $\phi_L$ to denote the area fraction of large particles and $\phi_s$ to denote the area fraction of small particles. For the system containing large particles only (i.e., $\phi_s = 0$), its radial distribution function (black line in Figure 3) demonstrated a typical gas-like structure with no peak in the *g(r)* plot.

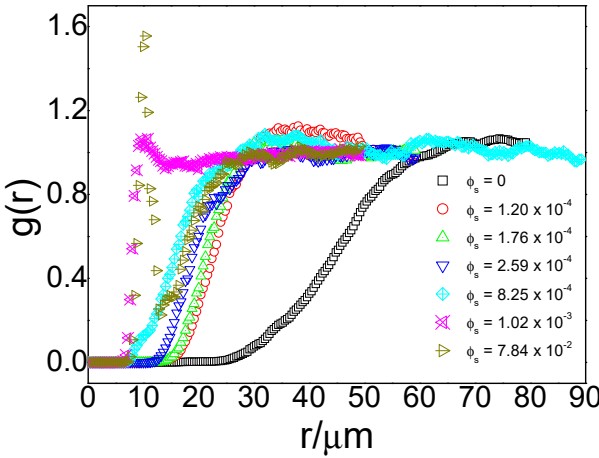

**Figure 3.** Experimental *g(r)* of 1-μm PS particles in binary-charged system with different small 0.21-μm PS particle concentration.

The nearest approachable distance between particles, i.e., the onset of nonzero *g(r)*, was approximately 25 μm. Same as our previous qualitative results in the binary system shown in Figure 2, narrowing between large particles in the presence of small particles appeared again in binary systems (colorful lines in Figure 3). With an increase in the concentration of small particles, the narrowing effects became stronger. For a binary system whose small-particle area fractions were around $10^{-4}$ (red line), the nearest approachable distance between large particles declined to approximately 15 μm. The *g(r)* maintained a gas-like structure, indicating that a random distribution of interfacial large particles remained in the presence of small particles. When the small-particle concentration increased to approximately $10^{-3}$ (aqua line), the particles could approach each other further down to less than 10 μm, and peaks began to appear in the *g(r)* curve at a distance of 10 μm (cyan line). When the particle concentration further increased to $7.84 \times 10^{-2}$, the peak at approximately 10 μm grew steeper. Peaks in *g(r)* curves of large particles only appeared in a system with a small-particle concentration higher than $10^{-3}$; no well-defined peaks were found in the other four binary systems. A peak in *g(r)* usually suggests an attraction at that distance.

### 3.2. U(r) of Particles of Different Concentrations

As we mentioned above, peaks in the PS particle curve indicated an attraction at that distance. To clarify this case, we compared the effective U(r) plot and treated small particles as the property of the medium to calculate the effective potential of large particles in the system using HNC approximation as shown in Figure 4. In general, the large particles in our unary and binary systems exhibited repulsion to each other. The repulsive range was long in the unary system as well as in the binary system with low small-particle density, each at a distance larger than 10 μm.

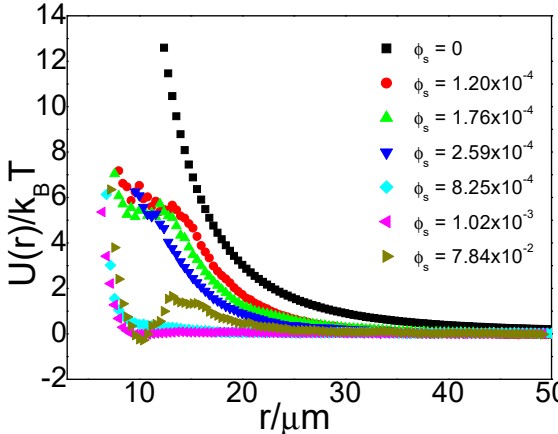

**Figure 4.** Pair-wise potential *U(r)* of 1-μm PS particles in binary-charged system with different small 0.21-μm PS particle concentration.

U(r) of unary particle system was consistent with our previous results [25], attraction between particle was not detected in our system. Just like Kralchevsky et al. have discussed [27,28], long-range attraction between interfacial particles are caused by interface deformation resulting from interfacial particles' weight, which is commonly found for interfacial system with particles larger than 5 μm [27]. In our system, interfacial polystyrene particles were 1 μm and 0.21 μm, and particle density is only 1.05 g/cm$^3$. These small particles are not heavy enough to deform interface to give attraction. Our previous study with 1 μm polystyrene at oil–water interface has shown that contact angle is 100° ± 7 and there is no deformation of interface from confocal image [25]. Contact angle of 0.21 μm polystyrene particle at oil–water interface is similar with 1 μm polystyrene [29], this smaller and lighter particle also cannot deform the interface to bring attraction between large particles.

For the binary system, with the presence of small particles, the repulsion between large particles weakened; when the small-particle concentration increased to $1.02 \times 10^{-3}$, an attractive well appeared at an approximate distance of 10 μm. At an even higher small-particle concentration of $7.84 \times 10^{-2}$, an attractive well of $-0.3\ k_BT$ was found.

### 3.3. Comparison with Published Work

Cui et al. [24] conducted a similar experimental study, where a suspension of silica (diameter $\sigma_L$ = 1.58 μm) and polystyrene (diameter $\sigma_S$ = 0.34 μm) was confined in a quasi-2D system by two plates, monolayer of large particle were sandwiched between this two plates. The resultant *g(r)* of large particles are presented in Figure A1, they use the large particles diameter $\sigma_L$ as a measure to evaluate the separation distance *r* between large particles. The area fraction ($N\sigma^2/A$) of large particles, $\phi_L$, at their confined layer was approximately 0.3, nearly 10 times our concentration. The concentration of small particles $\phi_s$ in their quasi-two-dimensional system was the volume fraction of the suspension, which ranged from 0, 0.005, 0.05, to 0.1. Small particle concentrations in Cui's system are at a higher level than our system.

In their results, well-defined initial peaks appeared in the *g(r)* of large particles at the distance around 1–1.5 times large particle diameter. Peak in *g(r)* appeared at a shorter distance with the increase in small-particle concentration $\phi_s$, and the nearest distance large particles could approach each other decrease accordingly with $\phi_s$ increase. For their unary large particle system, *U(r)* showed strong repulsion at around one particle diameter. In addition, the very short distance between confinement plates and large particles caused edge effects and resulted an attraction well right following repulsion branch in *U(r)* profile [24]. With $\phi_s$ increase in their quasi-2D systems, repulsion branch part in the pair-wise potential *U(r)* between large particles shifted to a shorter distance and the following attraction potential well became deeper (shown in Figure A1). Also, the minimum of the attraction

well shifted to smaller particle–particle separation. The attractive potential in their systems were approximately 0.3–0.8 $k_BT$. Although attraction between large particles by confinement inducing edge effects is absent in our system because we have very dilute particles of gas-like structure at interface, our systems shared one common feature with their systems: introduction of small particle in system made the particle–particle distance between large particles smaller. Small particles in Cui's quasi-2D system brought higher attraction between large particles while small particles in our decane-water interface system brought less repulsion between large particles. Note that Cui et al. reported nearly no measurable surface charge on particles; thus, the particles could essentially be considered an experimental hard sphere system. Volume exclusion between hard spheres naturally limits the nearest approachable separation between large particles to just around one large particle diameter, which accords with Cui's results. In contrast, the approachable distance between large particles in our system was much longer than one large particle diameter for the highly charged interfacial particles, providing longer-range (more than 10 times the particle size) repulsion than the hard sphere model.

### 3.4. Depletion Interaction

To interpret the attractive force in a binary particle system, Cui et al. compared it with an entropic force called depletion. Experimental observation of flocculation between particles in a binary colloidal suspension [30] was reported in 1925. Later, in 1958, Asakura and Oosawa [31] provided a detailed illustration of the mechanism of this attractive interaction. Depletion originates from volume exclusion effects in colloidal systems of binary sizes; a schematic illustration is provided in Figure 5. When considering the finite size of colloids and treating them as hard spheres, the small particles (depletants) are depleted from the large particle (with radius $R_l$) surface at a distance of the small-particle radius $R_s$; this zone is called the depletion layer. When two large particles get close and the surface-to-surface distance is less than the diameter ($\sigma_S = 2 R_s$) of the small particle, the exclusion of small particles will cause asymmetric distribution around one large particle, inducing an attraction. Depletion is an entropic interaction because the overlap of the depletion layer can provide the small particles more free space.

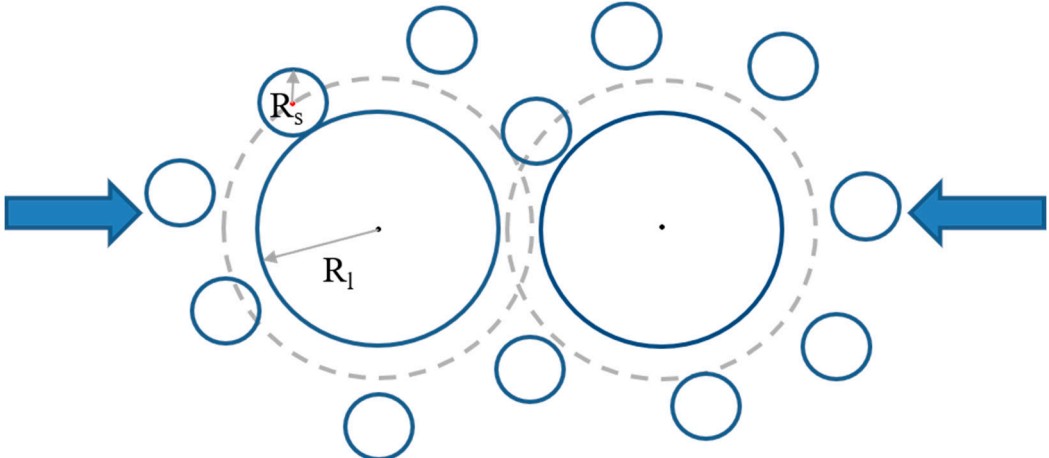

**Figure 5.** Schematic illustration of depletion attraction between hard spheres. Small particles with radius $R_s$ is depleted form the large particle surface, large particle radius is $R_l$. The ring around the large particle surface by dashed line is the depletion layer for small particles.

As depletion is caused by uneven surroundings of small particles around a large particle, the interaction range $r$ is $2R_l < r < 2(R_l + R_s)$ when the depletion layer outside the large particle overlaps. The interaction $U_o(r)$ is determined by the overlapped volume and concentration of small particles $\phi_s$. A calculation by the first order of small-particle concentration was provided by Asakura and Oosawa [31,32] as below when the depletant concentration is low:

$$U_0(r) = \phi_s \left[ \frac{4\pi}{3} \left( (R_l + R_s)^3 + \frac{r^3}{16} - \frac{3}{4} r(R_l + R_s)^2 \right) \right] \quad [2R_l < r < 2(R_l + R_s0] \tag{1}$$

$$U_0(r) = 0 \qquad [r > 2(R_l + R_s)] \tag{2}$$

Similarly, for the 2D case, we can replace the overlap volume term with the overlap area; the depletion between the hard disk is

$$U_0(r) = \phi_s \left[ 2(R_l + R_s)^2 \cos^{-1} \frac{r}{2(R_l + R_s)} - \frac{r}{2} \sqrt{4(R_l + R_s)^2 - r^2} \right] \quad [2R_l < r < 2(R_l + R_s)] \tag{3}$$

$$U_0(r) = 0 \qquad [r > 2(R_l + R_s)] \tag{4}$$

The depletion interaction range depends on the sizes of large and small particles, and the depth depends on the concentration of small particles. The interaction range of depletion matched the total effective potential range of 1–1.5 large particle size in the system from Cui et al.; however, in our system, this range was far below the measured result.

### 3.5. Structure of Binary Interfacial Particle System

The exclusion of particles to a distance far longer than their own sizes in our system appears clearly in the bright-field image (Figure 6). The black dots are large particles, and the shallow grey dots are small particles. Small particles were randomly distributed around large particles, except for system (f) with small-particle area fraction of $7.84 \times 10^{-2}$. In system (f), the *g(r)* showed a high peak at an approximate distance of 10 μm, whereas other distances displayed a distribution of large particles nearly identical to that in the system with lower small-particle densities (systems a–c). The interfacial structure in system (f) was mosaic under a high particle density: large particles, either individual or particle pairs, were surrounded by hexagonally ordered small particles. For large particle pairs, the separation between them was maintained around 10 μm, despite extreme crowding in the surroundings, concurring with the distance where the *g(r)* peak appeared.

We know that for charged interfacial particles, particles interact with each other by soft electrostatic potentials via both aqueous phase and oil phase. Charge clouds from particle's surface group dissociation in aqueous phase provide Coulombic electrostatic potential between interfacial particle, and this Coulombic potential is closely with the ionic environment, which can be characterized by Debye length $\kappa^{-1}$ in aqueous phase [33–35]. The counterpart electrostatic potential originates from charge on particle surface in the oil part [36,37], and range of electrostatic interaction via oil phase is much longer than the range of electrostatic interaction in aqueous phase due to oil's much lower dielectric constant. At near range when $\kappa r < 10$ ($r$ is the distance between particles), screened Coulombic repulsion dominated, while at far range $\kappa r \geq 10$, the counterpart electrostatic interaction dominated [35,38]. However, the charge on particle surface in oil part is unstable and resultant electrostatic interaction from oil phase decay with time to vanish [25,39]. In this work, we allow sufficient time for the charge on particle surface oil phase to vanish, thus we can skip discussion about electrostatic interaction from oil phase.

Because we used water as the aqueous sub-phase, the Debye length $\kappa^{-1}$ was approximately 1 μm in this ultra-low ionic medium. Thus, the screened Coulombic repulsion dominated in a range less than 10 μm in our systems, according with that large particles kept a distance around 10 μm in particle pairs under extreme uneven distribution of surrounding small particles in system (f). The individual particles in system (f) were nearly frozen by the surrounding small particles and maintained their initial separation when they were deposited at the interface. A depletion layer outside the large particles with an approximate length of 1 μm are apparent in the bright-field image(f). The freezing by small particles resulted in the *g(r)* in system (f) exhibiting a high peak at 10 μm; at the other separation, the particle distribution was nearly identical to the dilute systems (a–c).

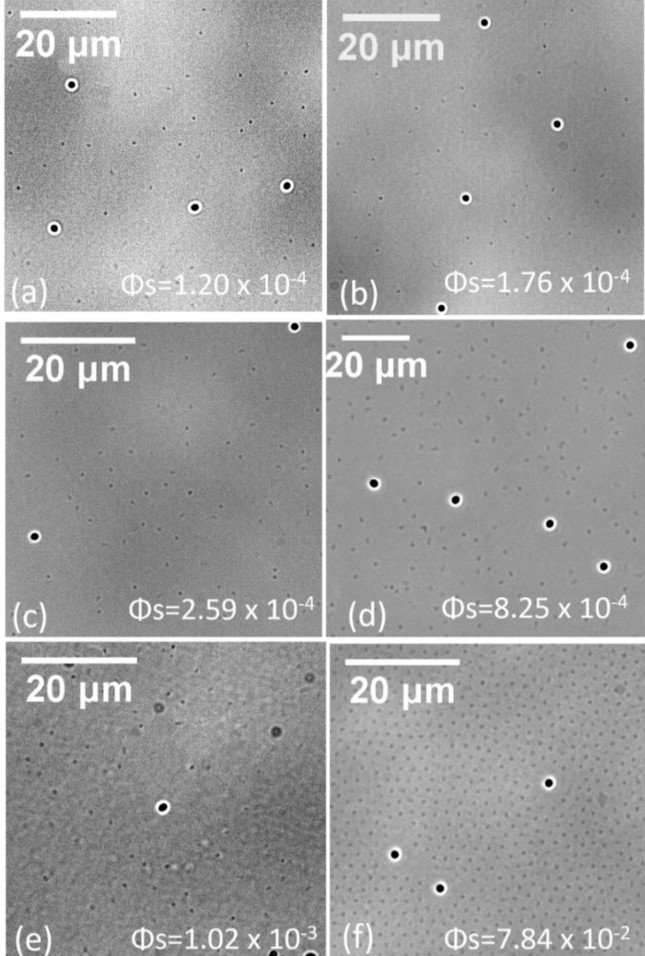

**Figure 6.** (**a**–**f**): Images of binary interfacial particles with a fixed concentration of large particles and increasing $\phi_s$, the concentration of small particles. The small particle structure changed from disordered to ordered as the concentration increased. Separation between large–large particles and large–small particles declined with an increase in the small particle concentration.

For systems (d–e), the concentrations of small particles were not high enough to freeze them; thus, only the statistical result of narrowing between the large particles is shown. At a low probability, the large particles were squeezed in the range of 10 μm where direct Coulombic repulsion between large particles dominated. For systems (a–c), the overall particle density was low, and substantial free space existed for small and large particles. Narrowing between the randomly distributed large particles was due to an increase in overall interfacial particles; as such, the *g(r)* of large particles continued to show a gas-like structure.

Previous reports have revealed that, for a 2D particle system with low area fraction [40] (less than 0.20), localization for the soft and hard disk was indistinguishable. The bright-field image of the binary system showed that the effective particle sizes that induced exclusion were beyond their actual size due to the negative surface charge. The particles were randomly distributed like gas in binary systems, with small-particle area fraction around $10^{-4}$ and showed nearly no compressive exclusion as seen in the bright image. These systems (a–c) were hence considered to be in a dilute situation. A detailed discussion follows to compare the soft particle with an effective hard disk of a radius of the onset of their *g(r)*. For the system with high density, since the area fraction $10^{-2}$ was within the area fraction limit of Foster et al.'s findings [40], the large particles acted like non-compressive hard particles of 10 μm in diameter.

### 3.6. Effective Interaction by Small Particles

The conventional depletion theory deals with infinite dilute particles immersed in a sea of small particles. However, in experimental systems, the concentration of large particles is dilute but finite. Other factors may also affect the results, such as the confinement of the system, which could induce unusual like-charge repulsion between particles in the unary particle system as reported by Cui et al. [24]. To understand the overall performance of small particles, we defined an effective potential reflecting the excess interaction by the presence of small particles: $\Delta U = U_{total} - U_{large}$, as Cui et al. did [24]. Our result is shown in Figure 7. The inset represents theoretical depletion by a comparably sized hard sphere as in the system by Cui et al. Details of Cui et al. 's result can be found in Figure A2.

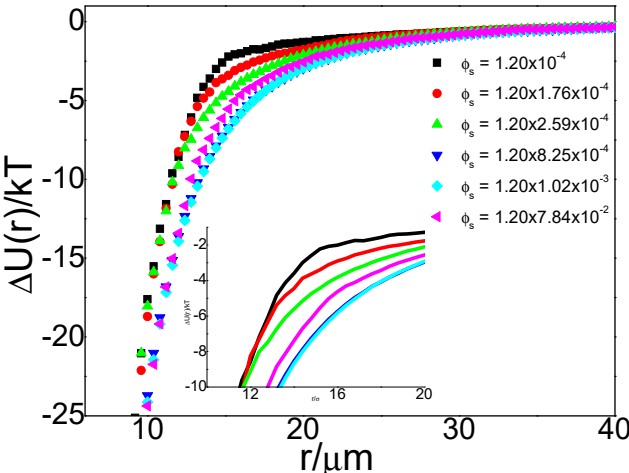

**Figure 7.** Effective interaction between large particles in charged interfacial systems: attraction range is longer by 20 times that of small particle sizes ($\sigma_S = 0.21$ μm), and attraction depth is larger than −25 $k_B T$, much longer range and stronger depth in interaction than in the uncharged system (details see Figure A2).

### 3.7. Experimental Results Vs. Theoretical Prediction

As noted earlier, the influencing range of a small particle's size effects at the interface in our system were far more than their physical size. Since for a small-area fraction, the hard particles and soft particles could be mapped to each other, we first used hard sphere mapping [40] to see the effects for a system with no crowding effects (systems a–c). Treating particles as hard sphere, we chose an effective diameter size of 15 μm (around the onset of *g(r)*) for large particles and a small-particle diameter of approximately 2 μm. We theoretically calculate the hard sphere depletion interaction $U_{dep}(r)$ and compare it with our experimental results, treating $\Delta U(r)$ as experimental $U_{dep}(r)$ (Figure 8). It is obvious that, even after adjusting the effective size to be much larger than the physical size of particles according to the interfacial particle image in Figure 6, the theoretical predicted attraction is still far below the experimental results in both range and depth.

Considering the stringent 2D confinement of oil–water interface of our system, we similarly compared our experimental results with the theoretical hard disk model. For the hard disk concentration, we calculated the average of the three dilute systems (systems a–c) as 0.007; results appear in Figure 9. The theoretical predicted results caused by the hard disk were even weaker than a hard sphere system. The range and depth of the theoretical prediction were each smaller than the experimental measured result. And the experimentally measured attraction was approximately 3 orders of magnitude stronger than the hard disk prediction.

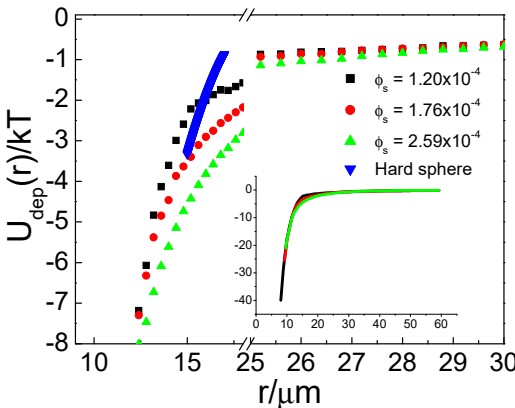

**Figure 8.** Comparison of theoretical hard sphere depletion and experimental results: attraction in experimental interfacial system was much larger. Effective size of particles $\sigma_L$ = 15 μm, $\sigma_s$ =2 μm.

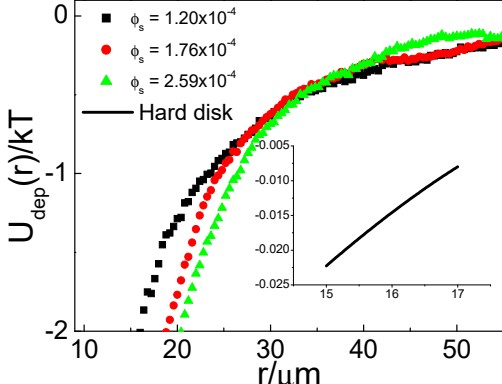

**Figure 9.** Comparison of theoretical hard disk depletion and experimental results: attraction in experimental interfacial system was much larger. Effective size of disk $\sigma_L$ = 15 μm, $\sigma_s$ =2 μm.

Although there has reported successful theoretical prediction for mapping between soft potential and hard potential in the ultra-dilute system [40], substantial divergence of experimentally measured results of soft interacting particles from hard particles were found in our results.

### 3.8. Electrostatic-Enhanced Interactions

Despite a report [41] suggesting that the softness of the potential can enhance depletion-like attraction between particles, few reported experimental results have supported this finding. Experiments [42] using a binary system of polystyrene particles measuring 100 nm and 25 nm in a water suspension with laser-light scattering have been reported. They found a depletion-like attraction enhanced by electrostatic repulsion, because of the enhanced excluded volume by the repulsion between the particle and the depletant. In low-electrolyte suspension, the interaction range was defined as $\sigma_L + \sigma_s + 2\kappa^{-1}$. However, the interaction range detected in our experiment (approximately 10 μm) was much longer than the incorporation of the long Debye length (1 + 0.21 + 2 = 3.21 μm). Additionally, the overall effective interaction between the large particles in their binary system was always attractive even under an ultra-low small particle concentration ($10^{-5}$). In our case, the overall effective interaction in the system remained repulsive. We identified an attraction enhanced by the particle charge, but more quantitative research remains to be done to elucidate the ultra-long range and depth of attraction with a minuscule concentration of small particles.

## 4. Conclusions

We reported an experimental result of interactions between large particles in binary systems containing large and small particles at a stringent 2D environment at oil–water interface, which has not been reported before to our knowledge. An ultra-long-range attraction between large particles caused by introduction of small particles to system was measured. Although a previous study [24] reported an enhanced attraction between large particles by small particle in a quasi-2D confined binary system with unknown mechanism, attraction caused by small particles in our stringent 2D system is longer in range and stronger in magnitude. We discussed the possible mechanisms for this anomalous attraction respective of hard sphere depletion in both 2D and 3D; electrostatic enhanced interactions were also discussed. However, no matching model or theory exists thus far regarding the mechanism of this long-range attraction. Further theoretical investigation and/or simulation regarding the electrostatic-enhanced depletion effects may clarify the structure and properties of a binary 2D charge system.

**Author Contributions:** Conceptualization, Z.Y. and T.N.; methodology, Z.Y.; software, Z.Y.; validation, Z.Y., and T.N.; formal analysis, Z.Y. and T.N.; investigation, Z.Y.; resources, T.N.; data curation, Z.Y. and T.N.; writing—original draft preparation, Z.Y.; writing—review and editing, Z.Y. and T.N.; visualization, Z.Y.; supervision, T.N.; project administration, T.N.; funding acquisition, T.N. All authors have read and agreed to the published version of the manuscript.

**Funding:** The financial support of this work by the Hong Kong Special Administration Region (HKSAR) General Research Fund (CUHK14304619, 2130642), and the Germany/Hong Kong Joint Research Scheme Sponsored by the Research Grants Council of Hong Kong and the German Academic Exchange Service of Germany (G-CUHK410/18).

**Acknowledgments:** The financial support of this work by the Hong Kong Special Administration Region (HKSAR) General Research Fund (CUHK14304619, 2130642), and the Germany/Hong Kong Joint Research Scheme Sponsored by the Research Grants Council of Hong Kong and the German Academic Exchange Service of Germany (G-CUHK410/18) is gratefully acknowledged.

**Conflicts of Interest:** The authors declare no conflict of interest. The funders had no role in the design of the study; in the collection, analyses, or interpretation of data; in the writing of the manuscript, or in the decision to publish the results.

## Appendix A

Reproduction of *g(r)* and *U(r)* of large particles in experimental binary quasi-2D suspension system by Cui et al. [24]: a suspension of silica (diameter $\sigma_L$ = 1.58 µm) and polystyrene (diameter $\sigma_S$ = 0.34 µm) was confined in a quasi-2D system by two plates. The resultant *g(r)* of large particles are presented in Figure A1, they use the large particles diameter $\sigma_L$ as a measure to evaluate the separation distance r between large particles.

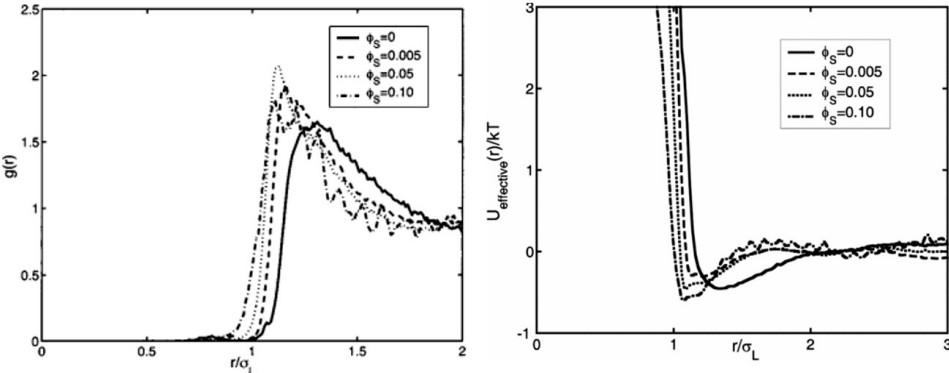

**Figure A1.** *g(r)* and *U(r)* of silica particles in non-charged binary colloid system: silica particles moved closer in binary system with attraction of 0.3–0.5 $k_B T$ at a distance of 1–1.5 with a large silica diameter.

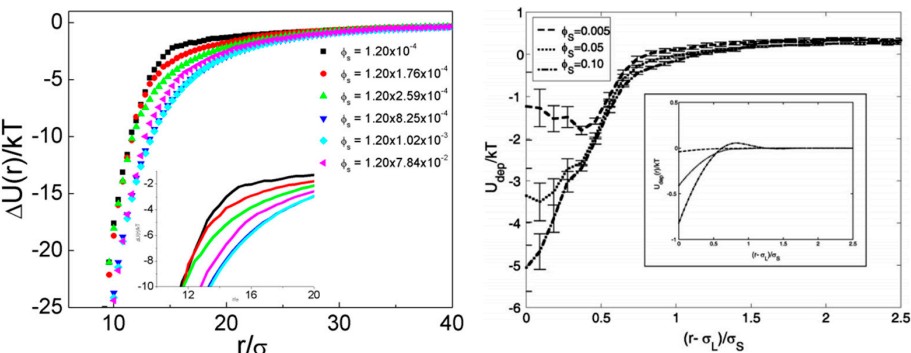

**Figure A2.** Comparison of effective interaction between large particles in charged (left): attraction range is longer than 20 times that of small particle size ($\sigma_S = 0.21$ μm), attraction depth is larger than $-25\,k_BT$. Cui et al.'s uncharged (right) interfacial systems: range of attraction is around 1~1.5 times small particle size, depth of the attraction caused by small particles is up to 5 $k_BT$. Charged system demonstrated a longer range and stronger depth in attraction than in the uncharged system.

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
