# Peer review of "Anomalous Long-Range Attraction in Colloidal Binary Mixtures at Fluid–Fluid Interfaces"

_colloids, doi:10.3390/colloids4030036_

Round 1
Reviewer 1 Report
Review of Anomalous Long-Range Attraction in Colloidal 3 Binary Mixtures at Fluid-Fluid Interfaces
By Zonglin Yi and To Ngai
The authors report measurements on the radial distribution function of particles at the water/decane interface. They investigate both unary and binary systems. Analysis of the binary systems reveals an attractive interaction between the larger particles that increases with the concentration of the smaller particles.
As with earlier studies in quasi 2D systems the authors have investigated depletion attraction to determine if it can explain the observed attraction. In this case the depletion interaction is too weak and of insufficient range to explain the observed attraction.
Comments
The authors work differs from previous work in that here the particles are trapped at the interface. The presence of an interface leads to additional forces that the authors appear not to have considered. Particles immersed at an interface between two phases are attracted to each other due to deformation of the interface[1] – in this case associated with the particle weight.
Whilst the particles here are small and the distances at which the attractive interaction is evident is large the magnitude of the attractive interaction is reasonable for spheres with an interfacial contact angle not equal to 90 degrees interacting through what Krachelvsky calls flotation forces.
Whilst the experiments are done carefully and the data is very interesting I think that the paper is not acceptable for publication until forces associated with interfacial curvature due to the presence of the particles at the interface are evaluated. I’d suggest that the increasing particle concentration of the smaller spheres leads to an increase in the curvature of the interface around the larger particle – and thereby increases the attraction.
I note also that the particles do not aggregate so a repulsion is required to balance the capillary attraction. The paper by Aveyard et al[2] describes how an electrostatic repulsion is possible by repulsion through the oil phase - this is longer in range than the repulsion through the aqueous phase because of the lower dielectric constant of the oil phase.
[1] P.A. Kralchevsky, K. Nagayama, Capillary interactions between particles bound to interfaces, liquid films and biomembranes, Adv. Colloid Interface Sci. 85(2-3) (2000) 145-192.
[2] R. Aveyard, B.P. Binks, J.H. Clint, P.D.I. Fletcher, T.S. Horozov, B. Neumann, V.N. Paunov, J. Annesley, S.W. Botchway, D. Nees, A.W. Parker, A.D. Ward, A.N. Burgess, Measurement of Long-Range Repulsive Forces between Charged Particles at an Oil-Water Interface, 88(24) (2002).
Reviewer 2 Report
The manuscript by Ngai et al reports experimental study of interactions between particles in 2D binary system. They found anomalous long range attraction in colloidal binary mixtures at oil-water interfaces. Their measurements showed that the enhanced attraction was stronger in 2D oil-water interface, increasing with the increase in concentration of small particles. Ultra-long-range attraction was found from a small amount of charged small particles. They compared the experimental results with the prediction based on hard disks or hard sphere, and the results can not be explained by volume exclusion effects. The manuscript provides a comprehensive introduction to the topic, and reports detailed and systematic experimental data, and careful discussion. I think this work will be very important to the field. So I recommend publication. Just a minor points.
- On page 8, the authors mentioned that water is the sub-phase, the Deby length was approximately 1 micron. This seems not appropriate, considering the particles are at the interface. The other phase is oil. Why don’t they choose the Deby length in oil medium, unless the authors can make sure the particles are only residing on water side? Clearly the Deby length will be much longer in oil.
- The manuscript is well-written overall. But somehow the conclusion part is terribly written. Very difficult to understand. Please revise it.
Round 2
Reviewer 1 Report
I am satisfied with the responses of the authors and now recommend publication